# High-resolution crystal structure of arthropod Eiger TNF suggests a mode of receptor engagement and altered surface charge within endosomes

Mattia Bertinelli [1], Guido C. Paesen[1], Jonathan M. Grimes [1,2] & Max Renner[1]

The tumour necrosis factor alpha (TNFα) superfamily of proteins are critical in numerous biological processes, such as in development and immunity. Eiger is the sole TNFα member described in arthropods such as in the important model organism *Drosophila*. To date there are no structural data on any Eiger protein. Here we present the structure of the TNF domain of Eiger from the fall armyworm *Spodoptera frugiperda* (SfEiger) to 1.7 Å from a serendipitously obtained crystal without prior knowledge of the protein sequence. Our structure confirms that canonical trimerization is conserved from ancestral TNFs and points towards a mode of receptor engagement. Furthermore, we observe numerous surface histidines on SfEiger, potentially acting as pH switches following internalization into endosomes. Our data contributes to the genome annotation of *S. frugiperda*, a voracious agricultural pest, and can serve as a basis for future structure-function investigations of the TNF system in related arthropods such as *Drosophila*.

[1] Division of Structural Biology, Wellcome Centre for Human Genetics, University of Oxford, 10 Roosevelt Drive, Oxford OX3 7BN, UK. [2] Diamond Light Source Ltd., Diamond House, Harwell Science and Innovation Campus, Didcot, Oxfordshire OX11 0DE, UK. Correspondence and requests for materials should be addressed to M.R. (email: maxrenner@strubi.ox.ac.uk)

Members of the tumour necrosis factor (TNF) alpha superfamily are critical in diverse cellular processes including tissue development, immune system regulation, proliferation, and apoptosis[1–3]. Most TNF superfamily members are type II transmembrane proteins with an extracellular portion containing a jelly-roll fold domain. They can exist either as membrane-anchored signalling molecules or as soluble proteins following release by proteolytic cleavage[4]. TNFα superfamily proteins exert their biological functions by forming trimeric oligomers that act as ligands for TNF receptors[5–9]. TNF receptor molecules bind the trimeric TNF ligand with between one and six extracellular cysteine-rich domains (CRDs), thereby activating signalling[4,10]. Typically, downstream signalling occurs by the recruitment of TNF-receptor-associated factors (TRAFs) or death domain proteins to the intracellular regions of the TNF receptors.

In humans at least 19 genes for TNF ligands and 29 genes for TNF receptors have been identified thus far[4]. This division of labour is thought to provide greater flexibility in the differential regulation of various physiological processes. However, in phylogenetically more ancient species, a smaller repertoire of TNF proteins is found, which is generally associated with a lower number of genome duplication events[11,12]. It is noteworthy that many vertebrate TNFs have essential functions in the adaptive immune system (such as in B cell and T cell regulation), yet the evolutionary origin of TNF ligands and receptors predates the development of adaptive immunity 500 million years ago. It is likely that genome duplications have expanded the number of TNFs which were then co-opted by the emerging vertebrate adaptive immune system[11]. Indeed, arthropods do not possess adaptive immunity, relying instead on innate immunity for host defence, and only a sole TNFα superfamily member, Eiger, has been identified in the *Drosophila* genome. Eiger may thus be considered an early evolutionary paralogue of vertebrate TNFα proteins. Eiger is a well-characterized and useful model in *Drosophila* as it allows TNF superfamily function to be studied uncoupled from adaptive immunity[13].

Like other TNF members, Eiger is a type II transmembrane protein with a single membrane-spanning helix, an intracellular portion, and an ectodomain harbouring the name-giving TNFα domain. Eiger has been shown to possess a broad variety of physiological functions, such as tissue development and tumour suppression[14]. Eiger activates the Jun N-terminal Kinase (JNK) signalling pathway by binding to its cognate TNF receptor. Two receptors of Eiger have been identified, termed Grindelwald and Wengen and which of these is utilized may be context and tissue specific[15,16]. After binding the receptor, dTRAFs (*Drosophila* TNF-receptor associated factors) are recruited and a sequential phosphorylation cascade is triggered ultimately leading to altered transcription[14]. Early experiments demonstrated that overexpression of Eiger induces cell death in the wing and eye[15,17]. In fact, Eiger is an important tumour suppressor which is involved in the elimination of polarity-deficient oncogenic epithelium[18]. While membrane-bound Eiger is a juxtacrine signalling molecule, acting only on directly contacting cells, it can also be cleaved from the cell by TNFα converting enzyme[19] and act at a distance, propagating the apoptotic signal further away and resulting in large groups of cells dying[20]. While this process is still not completely understood, it is thought to be important in tissue remodelling[21,22]. Interestingly, in certain cellular contexts such as in the presence of oncogenic Ras, Eiger can have tumour-promoting activity resulting in cellular overgrowth[23]. Furthermore, Eiger intervenes in the immune response by triggering the release of prophenoloxidase after the rupture of crystal cells[24], aiding in the clearance of extracellular pathogens[25], and facilitating the migration of macrophages[26]. Finally, Eiger is also involved in *sigmar*-induced autophagy, and in pain sensitization after UV damage in larvae by acting on sensory neurons[27,28].

While several mammalian members of the TNF superfamily have been structurally characterized, there is paucity in information on phylogenetically ancient paralogues. In this study we report the structure of the TNF domain of *Spodoptera frugiperda* Eiger (SfEiger) at 1.7 Å. A single crystal was obtained from which we could generate excellent electron density maps. Initially, using only the maps and the available Sf9 genome, we determined the sequence of SfEiger and inferred nucleotide positions of the exons. The structure confirms that canonical TNF trimerization is an ancestral property dating back to organisms with a single TNF. We observe a high concentration of histidine residues on the molecular surface of SfEiger, which may act as pH switches and have implications for signalling from endosomes. Finally, we report higher order assemblies within the crystal packing, which are consistent with receptor engagement and may possibly be related to the mode of accumulation of TNF components at the cell surface.

## Results

**The SfEiger TNF domain is coded by three exons**. We serendipitously grew a small crystal of SfEiger during the course of an unrelated structural study, from which we collected a complete data set to 1.7 Å. It was clear from the unit cell dimensions that this was not the originally intended target, but instead, in all likelihood, an endogenous protein of unknown sequence of our Sf9 expression system. It was thus necessary to devise a somewhat unorthodox strategy to solve and refine the structure. An overview of our adopted approach is shown schematically in Fig. 1a. We utilized the Wide Search Molecular Replacement server[29] to obtain initial phase information. The server carries out molecular replacement using approximately 100,000 SCOP domains as search models. We obtained a good solution with our diffraction data (LLG 65.000, PDBID: 1U5Y). We then employed SHELXE to perform automatic tracing and iterative density modification, yielding an essentially complete main chain and excellent density (Fig. 1b). The map features were clear enough to unambiguously identify the majority of side chains, allowing us to sequence by crystallography (Fig. 1c), yielding a model with ~60% of the amino acid sequence assigned.

We then utilized tBLASTn[30] to search for our partially assigned sequence against all six possible reading frames in the *S. frugiperda* genome[31]. We were able to infer three exons and the corresponding splice acceptor and donor sites delineating them in the Sf genome (Fig. 1d, Supplementary Fig. 1), covering our query and allowing us to complete the amino acid sequence of the TNF domain. We additionally confirmed the presence of the corresponding mRNA and validated the inferred sequence and splice sites by reverse transcribing SfEiger RNA from *S. frugiperda* RNA extracts and sequencing the product (Supplementary Fig. 2).

A multiple sequence alignment reveals the close relationship to TNF domains of related arthropods and confirms the identity of our protein as the Eiger homologue of *S. frugiperda* (Fig. 2). We concluded that our structure constitutes the soluble SfEiger TNF domain (part of the Eiger ectodomain) which has been released from Sf9 cells into the culture medium. Annotated homologues with over 85% sequence identity are from the tobacco cutworm *S. litura* (94% sequence identity), the cotton bollworm *H. armigera* (88% sequence identity), and the Kamehameha butterfly *V. tameamea* (86% sequence identity). For comparison, mammalian TNFα shows only 20% sequence identity to Sf Eiger.

**The structure of SfEiger**. Having determined the sequence of SfEiger, we then built and refined the structure (representative

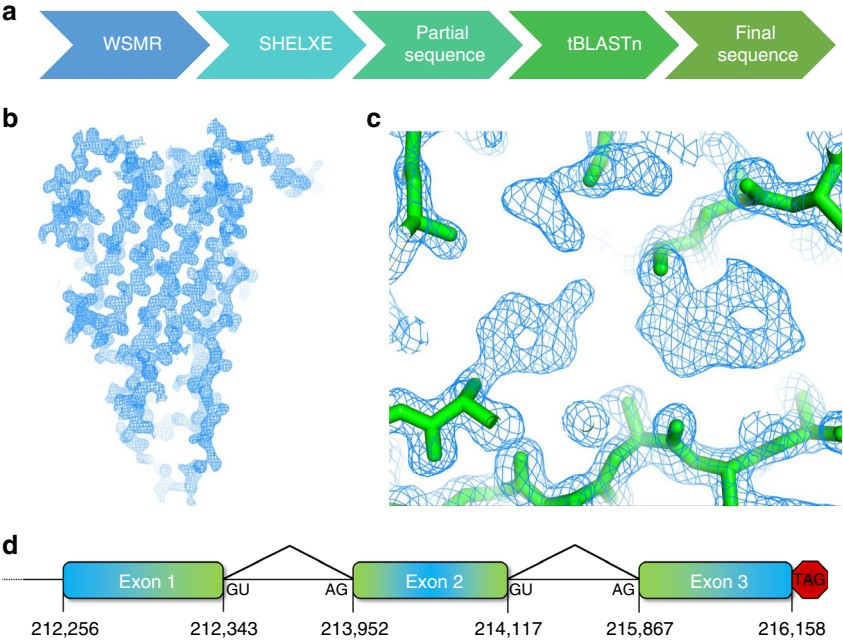

**Fig. 1** Determination of the SfEiger sequence. **a** Schematic overview of the workflow for the determination of the sequence and structure. **b** 2Fo–Fc electron density map contoured at 1.5σ after main-chain tracing with SHELXE and a single round of refinement. **c** Numerous side chains can be clearly identified (tyrosine on the left, tryptophan on the right) in the unbiased density maps. **d** Schematic representation of the three exons that code for the TNF alpha domain of SfEiger. The numbers under the boxes indicate the nucleotide positions at the beginning and at the end of each genomic sequence, respectively. Introns are indicated by angled lines

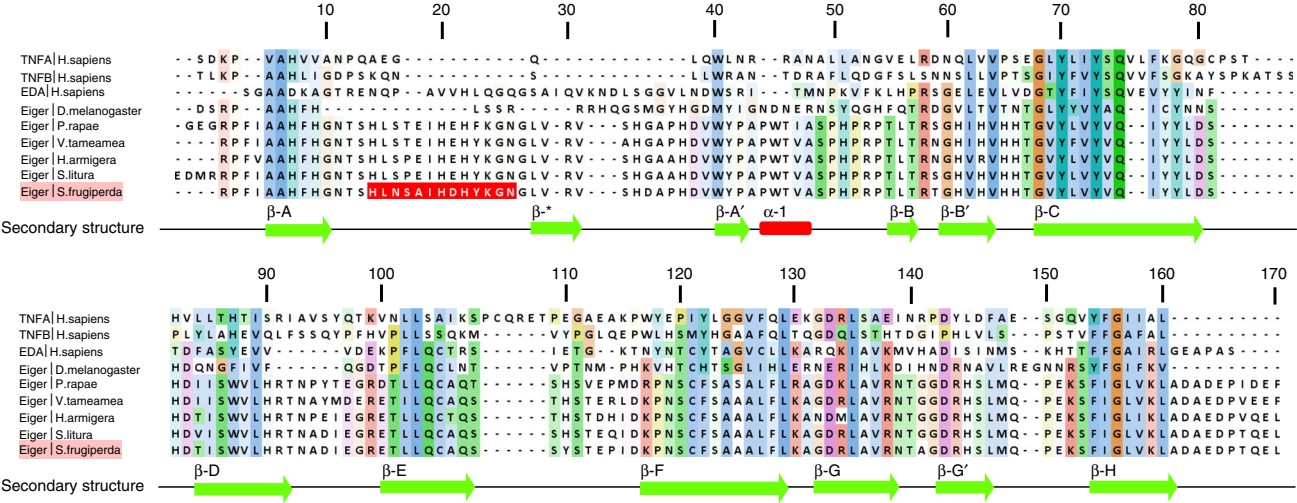

**Fig. 2** Multiple sequence alignment of select TNF members. Residues are numbered according to the sequence of the SfEiger TNF domain. Residues are coloured using the ClustalX colour scheme and secondary structure elements of SfEiger are indicated below the alignment. Residues highlighted in red are disordered in the structure but encoded in exon1

electron density is shown in Supplementary Fig. 3), obtaining excellent geometry and refinement statistics (Table 1). The TNF domain of SfEiger is composed of 170 amino acids with a short stretch of 13 amino acids not clearly visible in the density (indicated in Fig. 2), presumably due to flexibility. As the sequence of Eiger is not annotated in the Sf genome (and the intron–exon structure of the full transmembrane protein is undetermined) we decided to number the residues starting from position 1 of visible amino acids in the structure of the TNF domain. SfEiger adopts the jelly-roll fold shared by other members of the tumour necrosis factor (TNF) superfamily[6] (Fig. 3a). The domain is predominantly composed of β-strands organized

into two β-sheets. Secondary structure elements are indicated in Figs. 2 and 3a. A single disulfide bridge links strands E and F via residues Cys105 and Cys121.

We utilized the DALI server[32] to identify the closest homologue to SfEiger of known structure. The highest scoring hit retrieved by DALI was the human morphogenic signalling molecule Ectodysplasin A1 (EDA, PDBID: 1RJ7) with 22% sequence identity and 1.5 Å rmsd. EDA has repeatedly been predicted as the closest homologue to invertebrate TNFα superfamily members[11,17]. Comparison between the structures of SfEiger and human EDA (Fig. 3b) reveals some prominent differences. SfEiger possesses an additional α-helix located

**Table 1 Data collection and refinement statistics (molecular replacement)**

|  | *S. frugiperda* Eiger TNF domain |
| --- | --- |
| **Data collection** | |
| Space group | R 3:H |
| Cell dimensions | |
| $a, b, c$ (Å) | 55.2, 55.2, 142.1 |
| $\alpha, \beta, \gamma$ (°) | 90, 90, 120 |
| Resolution (Å) | 47.4–1.7 (1.72–1.69) |
| CC (1/2) | 1.0 (0.4) |
| $R_{merge}$ | 0.131 (0.974) |
| $I / \sigma I$ | 10.1 (1.2) |
| Completeness (%) | 97.5 (81.2) |
| Redundancy | 8.6 (3.5) |
| **Refinement** | |
| Resolution (Å) | 47.4–1.7 (1.72–1.69) |
| No. of reflections | 152050 (2675) |
| $R_{work}/R_{free}$ | 15.5/18.2 |
| No. of atoms | |
| Protein | 1240 |
| Water | 175 |
| *B*-factors | |
| Protein | 20.37 |
| Water | 37.33 |
| R.m.s. deviations | |
| Bond lengths (Å) | 0.005 |
| Bond angles (°) | 1.15 |

Statistics for the highest-resolution shell are shown in parentheses

between strands A′ and B which is absent in EDA (Fig. 3b, indicated by a black arrow). Furthermore, the loops connecting strands E and F and also D and E are notably longer in SfEiger (Fig. 3b, indicated by a white arrow). The 'DE flap' has been previously shown to be required for the formation of higher order assemblies of secreted B cell activating factor (BAFF) at neutral or basic pH[33,34].

We constructed a structural phylogenetic tree with the Structure Homology Program[35] using known structures of TNFα superfamily proteins. Such trees are often more sensitive than the sequence-based equivalents in identifying functional similarities of distant homologues[36]. SfEiger is not only positioned close to EDA (Fig. 3c) but also clusters with BAFF, APRIL, and TWEAK (TNF-related weak inducer of apoptosis). All of the latter proteins are unusual in that they bind to their TNF receptors via a single cysteine-rich domain which contacts a TNF monomer. In contrast, other TNF ligands recognize multiple cysteine-rich domains of their receptors, which bind across TNF dimer interfaces[37]. Consistently, the arthropod TNF receptors Wengen and Grindelwald also possess only a single cysteine-rich domain[11,22]. This suggests that TNF receptors with multiple CRDs appeared later in the evolution of components of the TNF system. Finally, we note that the disulfide bridge between strands E and F in Eiger is conserved in BAFF, APRIL, TWEAK, and EDA[8,38–40], while it is absent in more distant relatives like TRAIL or TNFα.

**The ancestral TNF Eiger forms a canonical trimer**. Previous structural studies on mammalian TNF ligands have demonstrated that all functional members form trimeric oligomers[4]. Although the SfEiger crystal contains a single protomer, as was also the case with TNFα family member TL1A[7], the threefold axis within the unit cell generates the canonical trimer, typical of TNF superfamily proteins[6–9]. The trimer resembles an inverted pyramid of ~60 Å height and ~50 Å width (Fig. 4a), with the β-sheet

composed of strands A, H, C, and F forming the majority of the oligomerization interface. The opposing B, B′, D, E, β-sheet faces the solvent. Additional contacts are formed between strands E and F of adjacent monomers. To experimentally confirm that SfEiger also forms a trimer in solution we cloned and recombinantly expressed this TNF domain. Comparison of the elution volume of SfEiger with a protein standard revealed a peak at higher molecular weight than the 44 kDa standard protein, consistent with a trimer and excluding a monomeric state (Fig. 4b, Supplementary Fig. 4).

The interface between the three monomers is dominated by hydrophobic interactions. Several highly conserved residues, specifically His7, Leu71, Tyr73, Phe122, Phe154, and Val158 form the inner core of the oligomerization region (Figs. 4c and 2). In addition, the single helix α1 contributes to oligomerization by packing against the C-terminal tail of a neighbouring protomer. Unlike TRAIL, no zinc atom can be found coordinated within the trimer, and the coordinating cysteines are not conserved in SfEiger[41]. Finally, the surface of the SfEiger trimer is decorated with a remarkable number of histidines (30 in total) (Fig. 4d). Besides being the likely reason SfEiger was copurified in the course of our nickel affinity step, these histidines may also confer a pH-driven change in surface charge distribution during the endocytic pathway of Eiger[18,42,43].

Next, we assessed the stability and dynamics of SfEiger. Analysis of the extent of the trimer interface using the PISA server[44] revealed a buried surface area of ~1930 Å² per protomer, which is comparable to other superfamily members such as TL1A (1977 Å²) or TRAIL (2261 Å²)[7]. Thermal denaturation analysis (Supplementary Fig. 5) revealed a melting temperature above 80 °C, indicating a highly stable protein. To further probe the conformational flexibility of the SfEiger trimer, we carried out 100 ns explicit-solvent molecular dynamics simulations (Fig. 5a, b). Average root mean square fluctuations were very low with values below 0.1 nm for the majority of residues (Fig. 5b). Some conformational flexibility could be observed in the DE loop (Fig. 5b, white arrow), and also in the A′A* loop (Fig. 5b, black arrow), consistent with the lack of clear electron density in the latter region in the crystal. Overall the data indicate that SfEiger forms a rigid and stable trimer in solution.

**Model of TNF receptor engagement**. The Eiger receptors Grindelwald and Wengen possess only a single cysteine-rich domain[11,22]. Single-CRD receptors essentially consist of a crucial and conserved β-hairpin which is followed by variable helix–loop–helix motifs[37]. The structural conservation (Fig. 3c) and the precedence of TNF ligand structures bound to receptors with only one CRD (i.e. BAFF and APRIL) allowed us to generate a coarse model of the engagement of trimeric Eiger with its receptor. We docked the BAFF-receptor onto SfEiger using the structure of the corresponding complex (PDBID: 4V46) by superimposing BAFF on SfEiger (Fig. 6a). Three copies of the receptor each bind to a monomer at the bottom of the inverted pyramid. In our model, the receptor engages primarily with the CD-loop, the GH-loop, and the D-strand of the SfEiger protomers. Notably, residues Asp83, Gln104, and Arg144 are found in close proximity to the receptor and are completely conserved throughout Eiger proteins, suggesting that they are important for receptor engagement (Figs. 6b and 2). In addition, one of the numerous surface histidines (His82) is located at the binding site and is also completely conserved in Eiger, indicating that it may serve a purpose in pH-dependent modulation of binding.

Clustering of TNF signalling components, including the respective receptors at the cell surface, and downstream interacting partners, is thought to enhance signal transduction

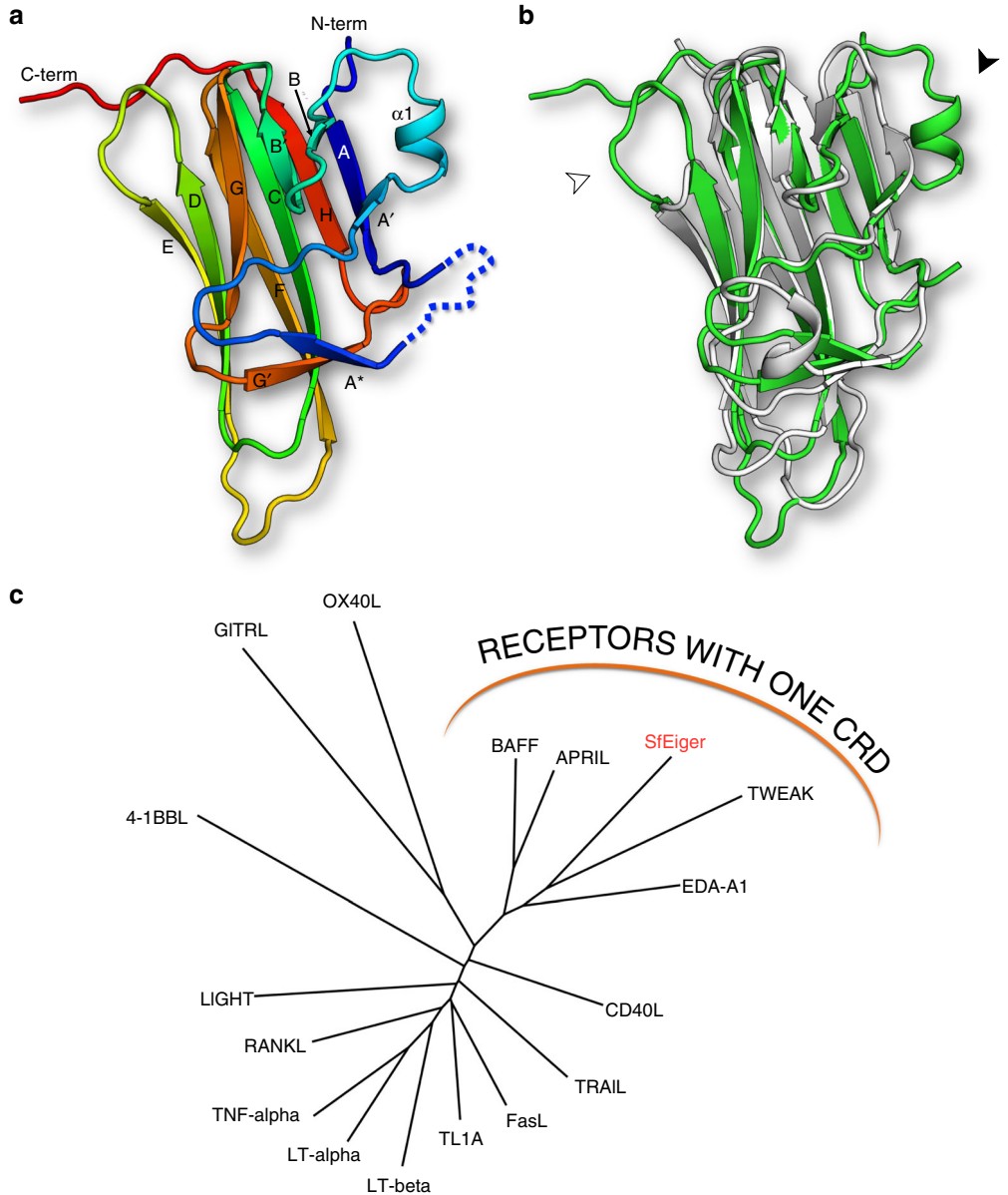

**Fig. 3** Structure of SfEiger. **a** Cartoon representation of monomeric Eiger coloured from N-terminus to C-terminus (blue to red, respectively). Secondary structure elements are labelled. **b** Comparison between the structure of SfEiger, coloured in green and of Ectodysplasin A, coloured in white (PDBID:1RJ7). The black arrow indicates helix α1, which sits at the interface with the neighbouring monomer. The white arrow indicates the extended DE loop. **c** Structure-based phylogenetic tree of the TNF superfamily. Family members that bind receptors with a single cysteine-rich domain (CRD) are indicated

and it has been suggested that hexagonal lattices form the basis of such clustering for several TNFα members[4]. We examined long-range crystal packing to assess if any information on possible higher order interactions between trimers may be found. Analysis of packing over multiple unit cells reveals sheets of SfEiger trimers which are aligned in a parallel fashion. Within the sheets the trimers are oriented so that their termini point in the same direction. These assemblies form a densely packed trigonal lattice (Fig. 6c). The lateral interactions are mediated by the DE loop and the C-terminal tail, which loosely contact neighbouring trimers. Within the sheets all three predicted binding surfaces for the Eiger receptor are available and receptors can be modelled on all sites without any major clashes (Fig. 6c, shown in purple). This is possible because the Eiger TNF receptors possess only one CRD, while for TNF receptors with large ectodomains containing multiple CRDs (as is the case for the majority of mammalian

TNFα superfamily members) such tight packing of trimers could not accommodate receptor engagement.

## Discussion

Signalling via the TNF superfamily plays essential roles in the regulation of development, immunity, and proliferation. TNF ligands may act either as membrane-bound forms or as soluble cytokines, activating signal transduction by binding to their respective receptors on the cell surface[4]. Because of their diverse functions in apoptosis and immunity, TNF ligands and receptors constitute attractive targets for intervention in autoimmune and cancer therapy. For instance, one therapeutic avenue makes use of specifically targeted apoptosis-inducing TNFs by coupling them to antibodies which recognize malignant cells[45]. The fundamental pathway of JNK signalling activation is conserved down to arthropods, where a single TNF, Eiger, has been identified and

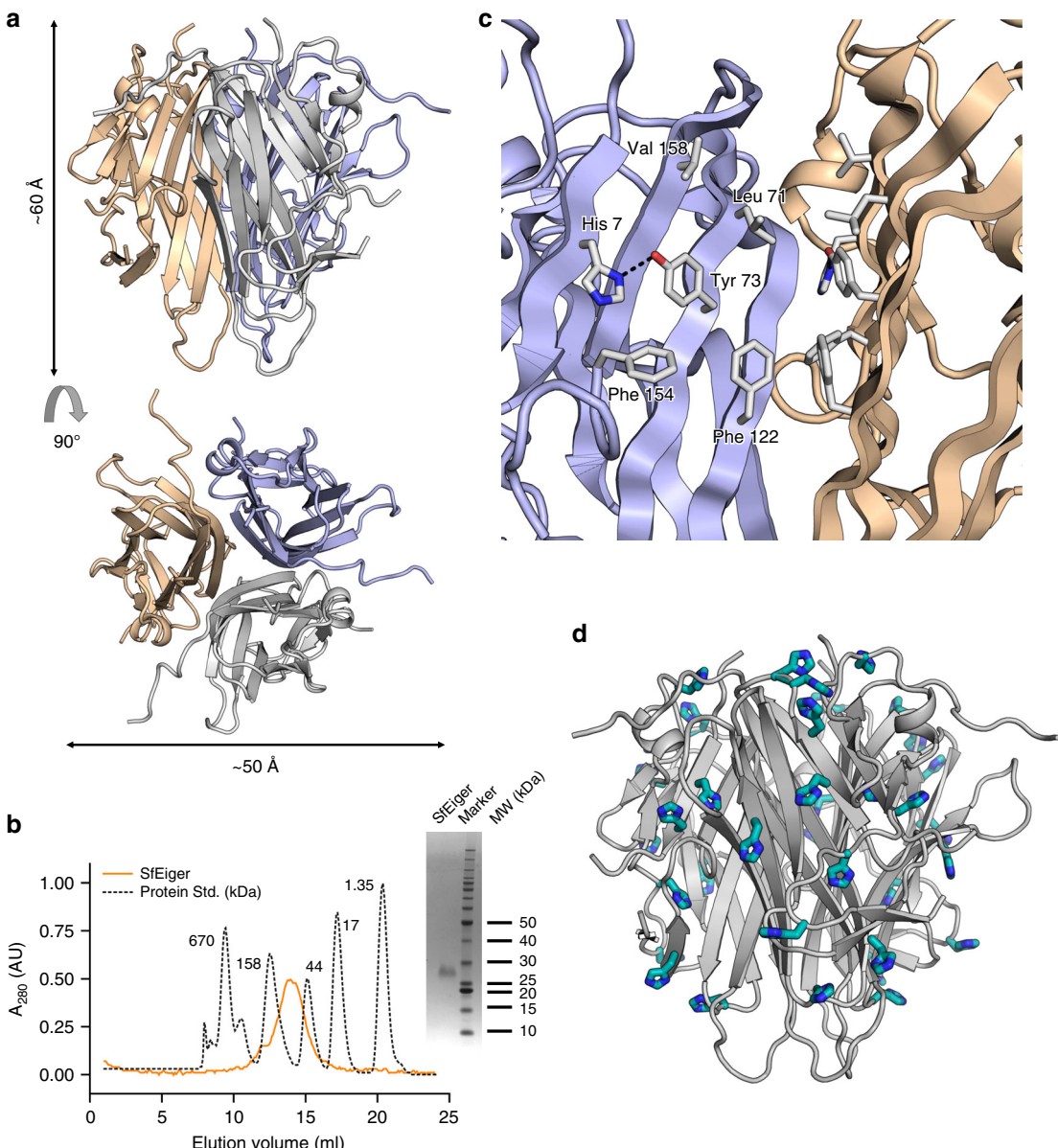

**Fig. 4** Structure of trimeric SfEiger. **a** Front- and top-view of the structure of trimeric Eiger. Double-sided arrows indicate the height and the width of the complex, respectively. **b** Size exclusion chromatography profile of recombinantly expressed SfEiger (orange). A protein standard is plotted in dotted lines for comparison. Molecular weights are indicated. The inset shows a representative SDS-PAGE of purified SfEiger. **c** Closeup of the interface between the three monomers of Eiger. Key residues forming the hydrophobic oligomerization interface are shown as sticks. Hydrogen bonds are indicated with a black dashed line. **d** Surface histidines on the assembled SfEiger trimer

has been widely studied in the *Drosophila* model organism[14]. We crystallized the ectodomain of endogenous Eiger of *S. frugiperda*, an important agricultural pest which impacts food security[46], and solved its structure. We were able to determine the sequence of the TNF domain of SfEiger utilizing the electron density. The crystal structure shows that the arthropod Eiger forms a canonical trimer which is also observed for distantly related mammalian TNFs, indicating that TNF superfamily and receptor trimerization has been conserved since before the phylogenetic split between protostomes and deuterostomes over 500 million years ago.

TNF signal transduction in mammals has been reported to be enhanced by clustering of multiple receptors at the cell surface and a model of a loose hexagonal lattice of TNF signalling components has been proposed recently[4]. In this model each TNF receptor trimer sits at a vertex of a hexagon and possesses three adjacent receptor trimers at the neighbouring vertices. The proposed assembly is compatible and able to reconcile the apparent symmetry mismatch between trimeric TNF receptors and the downstream TRAFs, which form dimers of trimers[47]. It should be noted that in the proposed loose lattice neighbouring TNF ligands would be too far separated to interact in the signalling state and clustering would be mainly mediated by TRAFs. However, TNF ligands have been previously reported to also interact with each other, as in the case of BAFF which is able to form biologically active, secreted virus-like 60-mers at neutral or basic pH, that can be observed from cell-lines which express BAFF endogenously[33,34]. This multimerization is dependent on the extended DE loop of BAFF[33].

Within the crystal packing of SfEiger, we observe sheets of tightly packed, laterally interacting, parallel trimers. The DE loop, which mediates assembly of BAFF 60-mers, is involved in contact

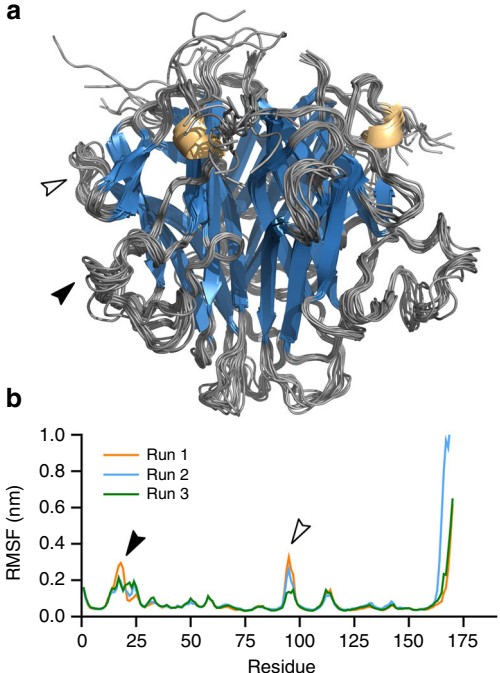

**Fig. 5** Molecular dynamics simulation of trimeric SfEiger. **a** Superposition of 10 conformers extracted at 10 ns intervals of the simulation. **b** Root mean square fluctuations (RMSF) of Cα atoms over the course of triplicate simulations, plotted per-residue. The black arrow indicates the position of the A′A*loop, and the white arrow indicates the DE loop

formation between the parallel copies in the crystal. It is tempting to speculate that the observed packing within the crystal may be related to the physiological state, and may hint at signal enhancement with high local concentrations of arthropod TNF components at the cell membrane. The observed trigonal TNF assembly is able to easily accommodate the single CRD of the short ectodomain of the receptor Grindelwald, and would be fully compatible with a trimer–trimer engagement for signalling activation (Fig. 6d). It may well be that a less densely packed, hexagonal lattice (indicated schematically in Fig. 6c) may occur. This would, in addition to improved binding strength by avidity through laterally interacting TNF ligands, also permit the formation of a network of TRAF dimers on the intracellular side. However, due to the inherent limitations of extrapolation from crystallographic packing, further study of TNF component accumulation at cell surfaces in arthropods is required.

Coordinated cell death through pro-apoptotic cytokines like Eiger is essential for healthy development. Apoptotic cells can proliferate a pro-death signal via a mechanism called apoptosis-induced apoptosis (AiA). A recent study on *Drosophila* wing discs demonstrated that expression of Eiger in apoptotic cells of the posterior compartment is able to induce non-autonomous apoptosis at long range in the neighbouring anterior compartment[21]. Similar mechanisms of regulated communal death in cell groups of multicellular organisms have been demonstrated to be important in a variety of organisms[20]. In the current study we crystallized endogenous SfEiger without any attempts at increasing its expression level. The observation that sufficient quantities of soluble Eiger, allowing for crystallization, were secreted into the culture medium by the Sf9 cells hints at a propagated pro-death signal. This might be due to the stress of overexpression of our target protein or, alternatively, a defence response to the infection with our recombinant baculovirus. Consistently, baculoviruses have

adapted the inhibitor of apoptosis protein p35, which is able to counteract apoptosis of the host cells by inhibiting required pro-apoptotic caspases[48].

Internalization of plasma membrane receptors into endosomes can have various functions in a given signalling pathway and signalling components can be assembled onto endosomes[49]. For instance, experiments using endocytosis inhibitors have shown that receptor mediated endocytosis is important for the activation of target genes by TNFα[50]. Furthermore, internalization is a requirement for signalling by TNF receptor 1 and subsequent formation of DISC (death-inducing signalling complex)[51,52]. Consistently, it has been demonstrated that *Drosophila* Eiger is transported to endosomes, where the JNK pathway is initiated[18,43]. Later studies identified Deltex to be important for the relocalization of Eiger to endosomes[53]. The surface of SfEiger is decorated with an astounding number of histidine residues (the second most abundant amino acid in SfEiger after alanine), the classical pH sensor in proteins. This includes the conserved His82 (corresponding to His342 in *Drosophila*) at the putative receptor binding site. Furthermore, *Drosophila* has an additional histidine (His394) located in the G–G′ loop, placing it directly in the centre of the receptor interaction surface. Histidines are well-known to have key functions in inducing conformational changes or altered interactions throughout the endocytic cycle, for instance during fusion of viruses. In the course of the endocytic pathway of Eiger, the accompanying shifts in pH may drastically alter the surface charge distribution of the protein, and it is likely that this modulates the affinity to interaction partners. However, more work is required to understand the significance of this.

## Methods

**Expression and purification.** Our intended target for this study was the envelope glycoprotein of the insect-specific Flavivirus Lammi virus[54], which we produce in Sf9 cells using a standard baculovirus expression system[55]. *Spodoptera frugiperda* Sf9 cells were cultured in SF-900 II serum-free media (Gibco) and infected with a recombinant baculovirus at a density of $1 \times 10^6$ cells/mL. After 72 h of growth, the culture was clarified by centrifugation ($1000 \times g$, 4 °C, 5 min). The supernatant was filtered through a 0.22 μm pore size filter and incubated with Ni Sepharose excel resin (GE Healthcare) overnight. After extensive washes with Buffer A (Tris base 20 mM pH 7.5, NaCl 1 M, Imidazole 20 mM) the protein was eluted with Buffer B (Tris 20 mM pH 7.5, NaCl 1 M, imidazole 500 mM). The eluted protein was then applied to a Superdex 200 column (GE Healthcare) for size exclusion chromatography. Fractions from the main peak were collected, incubated overnight with Peptide-*N*-Glycosidase F (PNGaseF), and concentrated to 3.8 mg/mL for crystallization screening.

Following the SfEiger crystal structure determination (see below), a synthetic gene for the TNF domain of SfEiger was ordered from GeneArt (ThermoFisher) and cloned into the pOPING vector using the In-Fusion ligation-independent cloning system (Takara)[56,57]. The vector containing the gene for SfEiger was used to generate a recombinant baculovirus following co-transfection with (flashBACULTRA) baculovirus (Oxford Expression Systems) using Cellfectin II (Invitrogen). Sf9 cells were infected with the baculovirus and after 72 h, cells were pelleted ($4000 \times g$, 10 min, 4 °C) and medium was recovered and filtered through a 0.22 μm filter.

The medium was then incubated overnight with Ni Sepharose Excel resin (GE Healthcare) and loaded onto a column. After extensive washing of the resin with Buffer A the protein was eluted with Buffer B. Subsequent steps of size exclusion chromatography were carried out using a Superdex 200 Increase 10/300 GL column (GE Healthcare).

**Reverse transcription coupled with PCR.** Sf9 cells infected with a baculovirus were pelleted ($1000 \times g$, 10 min, 4 °C), and used to obtain total RNA using the RNeasy Minikit (Qiagen) following the manufacturer's instructions. The RNA was then incubated with RQ1 RNAse-free DNAse (Promega) and used as a template for reverse transcription using SuperScript IV (ThermoFisher). The primer used for reverse transcription was 5′-GAGTCTACAGCTCTTGGGTAGGA-3′. The resulting cDNA was then amplified via PCR using Phusion polymerase (NEB). The PCR was carried out using the same reverse primer and the forward primer (5′-ACTT GGACTACCCCCAGACC-3′). The amplification product was verified by sequencing (Source Bioscience).

**Differential scanning fluorimetry.** Differential scanning fluorimetry experiments were performed with the Stratagene Mx3005P qPCR System (Agilent

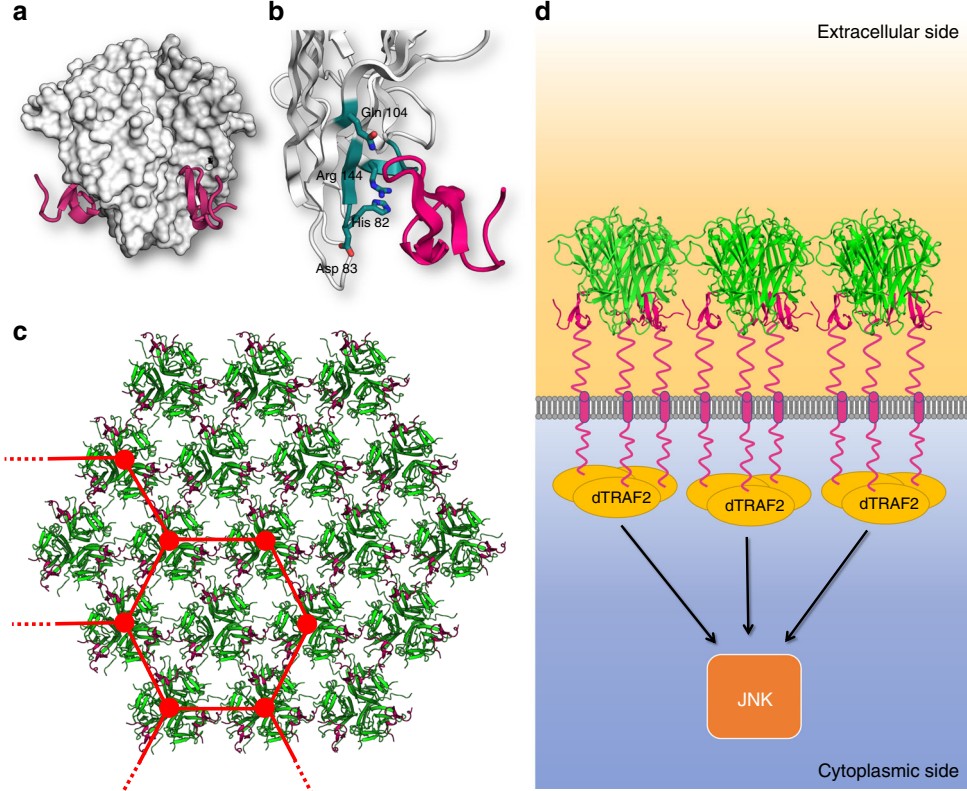

**Fig. 6** Model of receptor engagement of SfEiger. **a** Model of the interaction between SfEiger and a receptor. The SfEiger surface is coloured in white, while the receptor is coloured in purple and shown in cartoon representation. The single CRD was aligned onto SfEiger using PDBID 4V46. **b** Closeup of the putative region of interaction with the receptor. SfEiger is coloured white, the receptor in purple and conserved residues at the interaction interface are coloured in teal and shown as sticks. **c** View of a trigonal sheet of Eiger trimers (shown in green) related by crystallographic packing. The modelled receptor (purple) is included to show the absence of steric clashes. A hypothetical, sparser hexagonal lattice is indicated (see Discussion). **d** Model of laterally interacting SfEiger trimers, bound to receptors at the cell surface. The recruitment of dTRAF2 and downstream activation of JNK signalling is shown schematically

Technologies) using 2 μg of SfEiger. SYPRO orange dye (5000× stock; Invitrogen) was used at a final concentration of 3×.

**Molecular dynamics simulations.** In preparation for Molecular dynamics simulations the missing residues of SfEiger in the A'A* loop were modelled using SWISS-MODEL[58]. Molecular dynamics simulations and subsequent analyses were carried out in GROMACS v5[59]. The force field AMBER99SB-ILDNP* was utilized[60,61]. Prior to simulation, SfEiger was placed in a box of SPC/E water, with a distance of 1.0 nm from the simulation box edge. The GROMACS program genion was used to add 150 mM NaCl. Particle mesh Ewald summation was utilized to model long-range electrostatics[62], and the P-LINCS algorithm to restrain bond lengths. The simulation integration time step was 5 fs. Pressure and temperature were maintained at 1 atm and 300 K via the v-rescale thermostat and the Parrinello–Rahman barostat. After energy minimization and equilibration, three replicates were carried out for the simulation.

**Crystallization, data collection, and refinement.** Crystallization screening was performed by sitting drop using a Cartesian dispensing system[63]. A crystal of SfEiger appeared at 298 K after about 60 days in condition 1–26 of the MOR-PHEUS screen[64]. Crystals were cryoprotected with a solution of the mother liquor supplemented with glycerol (20% v/v) and cryocooled in liquid nitrogen. Diffraction experiments were performed using a wavelength of 0.9795 Å at 100 K. Diffraction data were collected at the I04 beamline at Diamond Light Source, Didcot, UK and processed with Xia2[65]. A suitable search model for molecular replacement was identified using the Wide Search Molecular Replacement server[29]. The best hit (PDBID: 1U5Y) was then used for molecular replacement in Phaser[66]. Based on the obtained solution, main-chain tracing was carried out using SHELXE[67], followed by several rounds of manual building with COOT[68] and refinement with REFMAC[69] and Phenix[70]. The structure was refined to an $R_{free}$ of 18.2% with 97.4% of residues in the Ramachandran favoured region and the remaining in the allowed region. Initially, in the absence of sequence information, we determined a partial protein sequence directly from the electron density. Using the partial sequence, we carried out BLAST searches against the non-redundant

protein sequences database, but were not able to find annotated SfEiger. We therefore determined the rest of the sequence from the Sf genome (see below). The model was validated with Molprobity[71], yielding a Molprobity score of 1.15. The structure factors and final coordinates have been deposited in the RCSB PDB with accession code 6I50.

**Protein identification.** The *Spodoptera frugiperda* genome[31] (GenBank ID: NJHR00000000.1) was downloaded from NCBI and converted to a compatible database using the makeblastdb command of BLAST[72]. BLAST searches were performed via Genome workbench (https://www.ncbi.nlm.nih.gov/ tools/gbench/). The partial sequence of SfEiger, derived from our crystallographic data, was used as query to search the *Spodoptera frugiperda* genome utilizing tblastn[30]. The resulting sequences of exons were then assembled and used to fully refine the crystallographic data. For sequence comparisons and to identify related Eiger proteins we queried the whole NCBI database of non-redundant protein sequences using BLAST.

**Structure and sequence analysis.** Sets of structural phylogenetic distances were calculated using the Structure Homology Program (SHP)[35], which were then converted to a phylogenetic tree using PHYLIP[73] and visualized with splitstree[74]. A full list of accession codes of used PDB entries is shown in Supplementary Table 1. Protein interfaces were analysed using the PISA server[44]. Structural images were generated using PyMOL (https://pymol.org). Multiple sequence alignments were performed with Clustal omega[75] and corresponding figures were prepared with Jalview[76].

**Reporting summary.** Further information on research design is available in the Nature Research Reporting Summary linked to this article.

## Data availability

Coordinates and structure factors have been deposited in the Protein Data Bank (http://www.rcsb.org/) with accession number 6I50. The size exclusion chromatography data can be found in Supplementary Data 1.

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

## Acknowledgements

This work was supported by the Wellcome Trust, UK. M.B. is part of the Wellcome Trust programme in Cellular and structural biology and is funded by Diamond Light source grant COL0108. M.R. is supported by a Wellcome Trust fellowship (204703/Z/16/Z), and the Wellcome Trust is also acknowledged for administrative support (203141/Z/16/Z). We thank Diamond Light Source for beamtime (proposal MX14774) and the staff of beamline I04 for assistance. In addition, we thank K. Harlos and T. Walter for assistance with crystallization and J. Keown for careful reading of the manuscript. We thank S. Wittmann for advice on reverse transcription.

## Author contributions

M.B., J.M.G., and M.R. conceived and designed research. M.B., G.C.P., and M.R. performed experiments. M.B. and M.R. analysed data. All authors contributed to the preparation of the manuscript.

## Additional information

**Competing interests:** The authors declare no competing interests.

