## [Peer Review File · Communications Biology]

Reviewers' comments:

Reviewer #1 (Remarks to the Author):

The manuscript 'Structure of the arthropod Eiger TNF suggests mechanistic details of signalling' by Dr. Bertinelli and colleagues presents the crystallographic structure of the TNF ligand Eiger from *Spodoptera frugiperda* (SfEiger) at 1.7 Å resolution. As reported by the authors, SfEiger was a contaminant in the sample of a different protein purified from insect cells, that crystallised serendipitously. The structure was solved by molecular replacement using the TNF April as a search model, which was identified by the Wise-Search-Molecular-Replacement program. The high-resolution diffraction data allowed full tracing of the initial electron density map, and revealed that the protein crystallised corresponded to SfEiger. Structure analysis suggests that SfEiger forms homo-trimers, with several surface-exposed histidine residues that may function as a pH-sensing switch during ligand internalisation. Comparison of SfEiger with known TNFs in complex with their cognate TNF-receptors suggested putative residues of SfEiger important for receptor binding (either Wengen or Grindelwald), and hinted at the possibility that the organisation of SfEiger trimers in the crystal lattice may reflect high-order Eiger/receptor complexes at the cell membrane.

The structure of *Spodoptera frugiperda* Eiger is the first structure of a TNF ligand from arthropods. The crystallographic protocols employed to determine the structure were well-executed, and all in silico analyses were conducted properly. However, the fact that the authors crystallised the protein serendipitously constitutes a great limitation in the insights that they can provide to scientists in the field, as there is no biochemical or functional validation of the molecular hypotheses formulated on the basis of the structural evidence (nor a purification protocol). Thus, in the lack of additional biochemical or functional evidence, I am wondering whether the manuscript is suited for Communications Biology.

Reviewer #2 (Remarks to the Author):

In this manuscript, Bertinelli etc. reported a serendipitously obtained crystal of an endogenous protein Eiger from the host cells sf9. The author determined the structure using the WSMR server and built the model based on the high-resolution density map combined with sf9 genome database. The structure of Eiger is unsurprisingly similar to its mammalian homologs. The crystal packing of Eiger indicates a trimeric conformation for receptor binding and an even higher order for downstream signaling. The author also built a binding model with the single CRD from its cognate receptors Grindelwald and Wengen, providing template for future structure-function investigations of the TNF system in related arthropods. The structure was well determined and analyzed, reflecting a strong background and technical on structure determination of authors. The paper was well written and the logic is reasonable, but my only concern is the biological significance of this study, e.g., if the author can prove the model of Eiger-CRD or how the histidines can affect the function/internalization experimentally. In another word, the current manuscript may fit better to some structural biology journals.

Reviewer #3 (Remarks to the Author):

Bertinelli et al., reported the structure of Eiger, ligand of the TNF family from SF9 insect cells, obtained serendipitously when they were targeting completely different object. It is a bizarre but interesting story with some novel discoveries: 1) this is the first structural report of TNF ligand from arthropods, in which they usually contains a solo TNF ligand. It should be of great interest for researchers to evaluate the original function of TNF family evolutionally. 2) The structure shows some interesting novel features such as rich content of histidines, and a potential function as pH

switches. 3) A strong evidence to support the novel interaction modes between TNF ligand and the single CRD of their cognate receptors as reported from that of BAFF and its receptor as well as 4) the potential clustering of TNF ligand to enhance the signal transduction found in BAFF. Both features seem evolutionally conserved from arthropods to human beings even after ~500 million years. However, to confirm the authenticity of the features, I like the authors to carry out the following experiments:

- 1) Change pH of the solution to confirm if Eiger forms cluster in vitro at some pH as BAFF does. And check if there are histidines within the DE loop, which could be sensitive to pH to form or disrupt hydrogen bonds between different trimers in solution at different pHs.
- 2) Similarly check if these clusters could be found in the insect cell extracts, this may be challenge due to lack of antibodies against Eiger currently. Alternatively, clone the gene and try to express it with tags.
- 3) The exact function of Eiger, instead of apoptosis, autophagy could be a main purpose. Instead of host defense, development could be more important.
- 4) The author may need to carry out a mass spectrum analysis to confirm the protein.

Gongyi Zhang,
National Jewish health

Response to Reviewers

We are grateful for the reviewers' encouraging comments and suggestions and we have considered these carefully. Many thanks for the opportunity to resubmit a revised manuscript. We believe that our updated version is significantly strengthened. Please find below point-by-point responses to the raised concerns.

Reviewer #1 (Remarks to the Author):

The manuscript 'Structure of the arthropod Eiger TNF suggests mechanistic details of signalling' by Dr. Bertinelli and colleagues presents the crystallographic structure of the TNF ligand Eiger from *Spodoptera frugiperda* (SfEiger) at 1.7 Å resolution. As reported by the authors, SfEiger was a contaminant in the sample of a different protein purified from insect cells, that crystallised serendipitously. The structure was solved by molecular replacement using the TNF April as a search model, which was identified by the Wise-Search-Molecular-Replacement program. The high-resolution diffraction data allowed full tracing of the initial electron density map, and revealed that the protein crystallised corresponded to SfEiger. Structure analysis suggests that SfEiger forms homo-trimers, with several surface-exposed histidine residues that may function as a PH-sensing switch during ligand internalisation. Comparison of SfEiger with known TNFs in complex with their cognate TNF-receptors suggested putative residues of SfEiger important for receptor binding (either Wengen or Grindelwald), and hinted at the possibility that the organisation of SfEiger trimers in the crystal lattice may reflect high-order Eiger/receptor complexes at the cell membrane.

The structure of *Spodoptera frugiperda* Eiger is the first structure of a TNF ligand from arthropods. The crystallographic protocols employed to determine the structure were well-executed, and all in silico analyses were conducted properly. However, the fact that the authors crystallised the protein serendipitously constitutes a great limitation in the insights that they can provide to scientists in the field, as there is no biochemical or functional validation of the molecular hypotheses formulated on the basis of the structural evidence (nor a purification protocol). Thus, in the lack of additional biochemical or functional evidence, I am wondering whether the manuscript is suited for Communications Biology.

We thank the reviewer for the positive comments and we acknowledge the lack of a reproducible purification protocol or biochemical characterisation for SfEiger. To remedy this we have ordered a synthetic gene of the SfEiger TNF domain, cloned it into a suitable vector, and carried out expression tests. While the expression levels were lower than

anticipated (perhaps limited by the inherent toxicity of expressing an insect TNF in insect cells), we were still able to produce and purify sufficient quantities of the His-tagged protein to carry out analytical size exclusion chromatography. Comparison to a protein standard shows that the elution volume of SfEiger is consistent with a trimeric state in solution (Figure 4B). In addition, we confirmed that the protein is highly stable and well folded by thermofluor (Supplementary Fig. 5). We have incorporated these additional biochemical data into the manuscript. Finally, to characterize the dynamics of trimeric SfEiger, we have carried out explicit-solvent molecular dynamics simulations, supporting the view of a highly rigid and stable trimer in solution (Figure 5). We think the inclusion of these experiments makes our study more well-rounded and the availability of a purification protocol should be useful for other researchers in the field.

While a detailed functional study of signalling and internalization of SfEiger would indeed be desirable, we feel this is out of the scope of our work and would significantly change the character of our story of the serendipitous crystallization of this arthropod TNF. Furthermore, we are limited by a lack of funding to pursue such a project. However, we note that there is abundant functional data on *Drosophila* Eiger and we think our work fits neatly into the body of available literature, which we reference in our MS. Our structure may also serve as a template to design future functional experiments in *Drosophila*. We thus believe that our characterization of an arthropod Eiger will be useful for a diverse audience beyond structural biologists and as such is suited for Communications Biology.

Reviewer #2 (Remarks to the Author):

In this manuscript, Bertinelli etc. reported a serendipitously obtained crystal of an endogenous protein Eiger from the host cells sf9. The author determined the structure using the WSMR server and built the model based on the high-resolution density map combined with sf9 genome database. The structure of Eiger is unsurprisingly similar to its mammalian homologs. The crystal packing of Eiger indicates a trimeric conformation for receptor binding and an even higher order for downstream signaling. The author also built a binding model with the single CRD from its cognate receptors Grindelwald and Wengen, providing template for future structure-function investigations of the TNF system in related arthropods. The structure was well determined and analyzed, reflecting a strong background and technical on structure determination of authors. The paper was well written and the logic is reasonable, but my only concern is the biological significance of this study, e.g., if the author can prove the model of Eiger-CRD or how the histidines can affect the function/internalization

experimentally. In another word, the current manuscript may fit better to some structural biology journals.

We thank the reviewer for the kind words on our structural analysis and manuscript. As noted above, we lack the funding to embark on a detailed functional investigation of this system and also believe that this would be well out of the scope of the story presented in our MS (although we do provide additional biochemical data for this revision). Functionally probing the interaction between Eiger and Grindelwald/Wengen and the subsequent internalization and signalling in a meaningful model system would constitute an entirely new study, well beyond our work here and requiring a significant investment of resources.

However, we do believe that our structure possesses multiple distinct features with implications for signalling and evolution which are of interest and use to a broader, biologically minded readership. We are confident in our Eiger-CRD model, based on precedence of corresponding mammalian complex architectures in the literature (reviewed in PMID 16914324). The work is also rather timely in light of the current outbreak of *S. frugiperda* (a pest causing billions of US\$ worth of crop losses annually, PMID 31102899) in 15 Chinese provinces, which threatens 10000s of hectares of crops (<https://www.fas.usda.gov/data/china-update-fall-armyworm-now-15-china-s-provinces>). For these reasons and the ones noted to reviewer 1 we believe that our molecular analysis of the apoptosis and inflammation related Eiger of an important pest will be of interest beyond the structural biology community.

Reviewer #3 (Remarks to the Author):

Bertinelli et al., reported the structure of Eiger, ligand of the TNF family from SF9 insect cells, obtained serendipitously when they were targeting completely different object. It is a bizarre but interesting story with some novel discoveries: 1) this is the first structural report of TNF ligand from arthropods, in which they usually contains a solo TNF ligand. It should be of great interest for researchers to evaluate the original function of TNF family evolutionally. 2) The structure shows some interesting novel features such as rich content of histidines, and a potential function as pH switches. 3) A strong evidence to support the novel interaction modes between TNF ligand and the single CRD of their cognate receptors as reported from that of BAFF and its receptor as well as 4) the potential clustering of TNF ligand to enhance the signal transduction found in BAFF. Both features seem evolutionally conserved from arthropods to human beings even after ~500 million years.

However, to confirm the authenticity of the features, I like the authors to carry out the following experiments:

1) Change pH of the solution to confirm if Eiger forms cluster *in vitro* at some pH as BAFF does. And check if there are histidines within the DE loop, which could be sensitive to pH to form or disrupt hydrogen bonds between different trimers in solution at different pHs.

As described to reviewer 1 we have ordered the synthetic gene of SfEiger, cloned it, and purified the protein. To address this point we have carried out analytical size exclusion chromatography (SEC) at neutral pH (as would be the condition at the cell surface). The elution volume of SfEiger is consistent with a trimer (Figure 4B). As stated by the reviewer, secreted mammalian B-cell activating factor (BAFF) has been observed to form virus-like 60-mers at neutral pH or basic pH 9.0 (PMID 16475789). We ran an additional SEC at the latter pH revealing an only very slight shift of elution volume, still indicating a trimeric Eiger, with no evidence of higher order clustering (Supplementary Fig. 4). These results are now included in the MS. There is no Histidine residue present in the DE loop of SfEiger.

To clarify, we do not suggest that SfEiger forms higher order oligomers in a pH dependent manner as is the case with BAFF. We have made this more clear in the manuscript text now. Instead, we speculate that the lateral packing arrangement in the crystal may reflect conditions of high local concentrations in two dimensions of receptor and ligand as is the case at the cell membrane (as correctly noted by Reviewer 1). We thus do not expect to observe this *in vitro*. However, we do acknowledge the limitations of extrapolating from crystallographic packing and have toned down the text in this regard.

2) Similarly check if these clusters could be found in the insect cell extracts, this may be challenge due to lack of antibodies against Eiger currently. Alternatively, clone the gene and try to express it with tags.

As described above we expressed tagged SfEiger and could not detect higher-order oligomerization *in vitro*.

3) The exact function of Eiger, instead of apoptosis, autophagy could be a main purpose. Instead of host defense, development could be more important.

Indeed, regulating development is one of the main functions of Eiger. We have further emphasized this in the Introduction. There is also evidence for Eiger being involved in autophagy. For instance, in PMID 25836674 the authors demonstrate that the expression of TIPE family member *sigmar* is regulated by Eiger and modulates autophagy in larval salivary glands. We now reference this research in our MS.

4) The author may need to carry out a mass spectrum analysis to confirm the protein.

As we only obtained a single, tiny crystal of SfEiger, which we could not retain after data collection, we unfortunately have no material to perform mass spectrometry on. However, to address this point, we instead generated cDNA from total RNA extracts of *S. frugiperda* utilizing primers specific to Eiger mRNA (Supplementary Fig. 2). We obtained a clear main product consistent with the expected amplicon size and we sent this for sequencing. The sequencing results confirm that SfEiger is expressed and the amino acid sequence matches the one we predicted perfectly (Supplementary Fig. 2C). In addition, we would like to note that there is a sole TNF in *S. frugiperda* whose sequence was used to refine our crystal structure. The crystallographic statistics are excellent and R_{free} values of 0.18 (as is the case here) would generally be absolutely unachievable if we had used the sequence of an incorrect protein for refinement.

REVIEWERS' COMMENTS:

Reviewer #3 (Remarks to the Author):

The authors addressed all questions raised by others and me.